# Chimeric Antigen Receptor Design and Efficacy in Ovarian Cancer Treatment

**DOI:** 10.3390/ijms22073495

**Published:** 2021-03-28

**Authors:** Katarzyna M. Terlikowska, Bożena Dobrzycka, Sławomir J. Terlikowski

**Affiliations:** 1Department of Food Biotechnology, Medical University of Bialystok, Szpitalna 37 Street, 15-295 Bialystok, Poland; kterlikowska@gmail.com; 2Department of Gynaecology and Obstetrics, Medical University of Bialystok, M. Sklodowskiej-Curie 24A Street, 15-089 Bialystok, Poland; bdobrzycka@gmail.com; 3Department of Obstetrics, Gynaecology and Maternity Care, Medical University of Bialystok, Szpitalna 37 Street, 15-295 Bialystok, Poland

**Keywords:** chimeric antigen receptors, ovarian cancer, relapse, chemoresistance

## Abstract

Our increased understanding of tumour biology gained over the last few years has led to the development of targeted molecular therapies, e.g., vascular endothelial growth factor A (VEGF-A) antagonists, poly[ADP-ribose] polymerase 1 (PARP1) inhibitors in hereditary breast and ovarian cancer syndrome (*BRCA1* and *BRCA2* mutants), increasing survival and improving the quality of life. However, the majority of ovarian cancer (OC) patients still do not have access to targeted molecular therapies that would be capable of controlling their disease, especially resistant or relapsed. Chimeric antigen receptors (CARs) are recombinant receptor constructs located on T lymphocytes or other immune cells that change its specificity and functions. Therefore, in a search for a successful solid tumour therapy using CARs the specific cell surface antigens identification is crucial. Numerous in vitro and in vivo studies, as well as studies on humans, prove that targeting overexpressed molecules, such as mucin 16 (MUC16), annexin 2 (ANXA2), receptor tyrosine-protein kinase erbB-2 (HER2/neu) causes high tumour cells toxicity and decreased tumour burden. CARs are well tolerated, side effects are minimal and they inhibit disease progression. However, as OC is heterogenic in its nature with high mutation diversity and overexpression of different receptors, there is a need to consider an individual approach to treat this type of cancer. In this publication, we would like to present the history and status of therapies involving the CAR T cells in treatment of OC tumours, suggest potential T cell-intrinsic determinants of response and resistance as well as present extrinsic factors impacting the success of this approach.

## 1. Introduction

Ovarian cancer (OC) is the 8th most common form of cancer in women worldwide with an estimated 295,414 new cases and 184,799 deaths annually. It has the worst prognosis and the highest mortality rate among gynaecological cancers [1]. Moreover, it is predicted that by the year 2040 the mortality rate of this specific type of cancer will rise significantly [2,3]. OC develops asymptomatic, and there is no proper screening program that would facilitate early-stage diagnosis [4,5]. OC metastasis occurs remarkably early in the disease development process. Tumour cells extrude from the primary tumour, survive anchorage-independent apoptosis as free-floating cells or form spheroids, then spread across the peritoneal cavity where they proliferate and interact with mesothelial cells and adipocytes of the omentum. Due to the insidious nature of this disease, most patients with OC are diagnosed with advanced stages of the disease mainly due to intraperitoneal spread and often the presence of distant metastases (International Federation of Gynaecology and Obstetrics, FIGO stage III/IV disease) [6]. The early detection of OC remains challenging because clinically apparent symptoms only manifest during the disease’s later stages. Metastases are associated with a poor prognosis where the typical overall survival rate ranges from weeks to months if untreated [7]. Only 15% of patients are diagnosed at an early stage, whereas the majority of women are diagnosed with metastatic cancers (92% vs 29% 5-year survival rate) [8]. Patients diagnosed with stage III or IV OC have a 5-year survival rate of less than 25%, including aggressive surgical resection and first-line chemotherapy drugs administration [9]. Therefore, conventional treatments such as debulking surgery and combination chemotherapy are rarely able to control the progression of the tumour, and relapses are frequent. Although up to 75% of patients achieve a good clinical response following initial therapy, almost all will ultimately relapse and eventually develop the chemotherapy-refractory disease. Consequently, the OC survival rate has not changed significantly despite decades of research [10]. Therefore, we need novel and effective therapeutic methods that would ensure beneficial long-term clinical outcomes for patients with OC. Metastasis from OC can occur via the transcoelomic, haematogenous, or lymphatic route. Transcoelomic metastasis being the most common is responsible for the highest morbidity and mortality rates among women with OC [11,12]. Malignant epithelial tumours account for 90% of all OC cases. Histopathology, immunohistochemistry and molecular genetic analysis are used to perform classification [13,14,15]. In samples obtained from patients, it is possible to distinguish high-grade serous carcinomas (HGSC), endometrioid carcinomas (EC), clear cell carcinomas (CCC), mucinous carcinomas (MC) and low-grade serous carcinoma (LGSC) [16,17]. Over two-thirds of OC cases account for HGSC. Immune signatures define a subgroup of HGSCs with a high percentage of infiltrating lymphocytes that have better survival outcomes. On the other hand, reactive stromal signatures with high levels of desmoplasmia, activated myofibroblasts, vascular endothelial cells and extracellular matrix remodelling is an indicator of the poorest prognosis [18,19,20].

OC is responsible for the dysregulation of the immune system in a multistep cooperative process. Stimulating the host to initiate the immune response against tumours requires the following: (1) a sufficient amount of effector T cells must be produced in the body to recognise tumour antigens effectively; (2) these cells must identify, present and infiltrate tumour tissue; (3) must overcome the inhibition of the tumour microenvironment (TME) on the immune network; (4) must directly identify tumour antigens and kill tumour cells; and (5) must maintain the activity of anti-tumour T cells for a long time [21,22]. While the TME tumour-associated immune cells may be initially involved in restricting tumour growth, these cells are also immunosuppressive and contribute to tumour progression due to their ability to block the host anti-tumour responses and drive the angiogenesis of the tumour. [23]. Myeloid leukocytes are the main components of the immune system supporting tumour expansion through secretion of growth factors, inhibition of anti-tumour T cells via the production of arginase and vascularisation [24]. For tumour growth and cancer dissemination tumour fibroblasts are responsible, while regulatory T cells cause immunosuppression of the host’s system [25]. Tumour-associated macrophages (TAMs) adopt an alternative phenotype M2, characterised by enhanced tissue regenerative responses and local immune suppression [26]. OC cells secrete large amounts of IL-10, promoting differentiation of dendritic cells DC to CD14+CD1 a macrophage-like cell with reduced T-cell activation properties [27]. Although studies regarding immune cell profiles by histologic subtype are limited, researchers found that HGSC had the highest number of tissue cores stained with the pan leukocyte marker, CD45 and also more frequently FoxP3, CD25 or CD20 compared to other subtypes. Tumours with endometrioid histology (EC) had the second-highest and clear cell (CCC), as well as mucinous (MC), had the lowest percentages with infiltrates overall [28]. One mechanism by which several different types of immune cells are suppressed in the TME is through the production of indoleamine 2,3-dioxygenase (IDO) [29].

Currently, the first-line treatment regimen for OC patients is complete debulking surgery. Despite the fact that this type of surgery constitutes the basis for OC treatment, it is rarely sufficient alone for patients with advanced disease and must be combined with chemotherapy [30]. Increased understanding of OC biology and chemoresistance gained over the last few years led to the development of targeted molecular therapies improving the survival and increasing the quality of life in OC patients (VEGF-A antagonists, PARP inhibitors in *BRCA1* and *BRCA2* mutants). On the other hand, the majority of OC patients still do not have access to targeted molecular therapies that would be capable of controlling their disease [31]. One of the promising strategies overcoming non-specific activity and disease relapse is immunology engineering. Cell-based cancer immunotherapy represents a promising option for patients without access to treatment alternatives. This approach focuses on the use of the patient’s immune system to destroy the OC cells and ideally on triggering an immunological memory response.

## 2. What Is CAR?

Chimeric antigen receptors (CARs) are recombinant antigen receptors located on T lymphocytes or other immune cells that redirect their specificity and functions [32]. The moieties used to bind to antigen fall in three general categories: (a) single-chain variable fragment (scFv) derived from antibodies; (b) antigen-binding fragment (Fab) selected from libraries or (c) nature ligands that engage their cognate receptor. The main rationale behind the use of CAR receptors in cancer immunotherapy is the rapid production of tumour-targeting T cells, bypassing the barriers and incremental kinetics of active immunisation [33]. The CAR-modified T cells acquire unique properties and act as ‘living drugs’ that may result in short-term, as well as long-term effects [34]. There are four generations of CARs used in clinical practice. The core structure of all four generations is an extracellular antigen recognition region with scFv, which is responsible for immunogenicity, affinity and specificity [35]. With scFvs, CARs can target specific cells and trigger downstream signals. Fragments of scFvs derive from an antigen-specific monoclonal antibody (mAb) [36]. The receptor’s extracellular domain originates from a cluster of differentiation CD4 and CD8. The transmembrane domain is usually derived from CD8, CD3-Ϛ (zeta), CD28 and intracellular tail including members of the tumour necrosis factor (TNF) receptor family, 4-1BB (CD137), OX-40 and CD27, has been incorporated to second and third generation [37]. The fourth generation of CARs is also called TRUCK T cells and was engineered to induce cytokines production, for example, IL-2, IL-12, IL-15 or granulocyte-macrophage colony-stimulating factor (GM-CSF) [38]. The green fluorescent protein (GFP) is a protein that exhibits bright green fluorescence when exposed to light in the blue to ultraviolet range. It can be added to every generation of CAR in term to estimate its specificity to bind target antigen via fluorescence microscope. Figure 1 represents the structure of CARs.

Eshhar et al. designed structures that specifically recognise and respond to the antigen without signalisation of major histocompatibility complex (MHC) [39]. Unfortunately, first-generation CARs proved to be of limited clinical benefit because of failure in directing T-cell expansion upon repeated exposure to the antigen [40]. The 4-1BB ligand, CD137L is found on APCs (antigen-presenting cells) and binds to the 4-1BB superfamily, which is expressed on activated T Lymphocytes [41]. Savoldo et al. proposed incorporation of one stimulatory domain CD28 or 41BB to the second-generation CARs [42]. Third-generation CARs were formed by the incorporation of two or more costimulatory domains. On the other hand, their clinical effect in comparison to second-generation remains controversial [43,44]. The fourth-generation was developed to redirect T cells for universal cytokine killing, via the addition of an IL-12 expression cassette. IL-12 can accumulate in the target tissue and recruit a second wave of immune cells, e.g., NK cells, macrophages [45,46].

## 3. How Are CARs Engineered?

Scientists use several gene transfer methods to insert a specific gene into mice or human T lymphocytes. These methods differ in the expression levels and stability of mentioned CAR-T cells. In general, there are two main approaches in immune engineering: viral and non-viral [47]. Viral vectors have high infection rates; however, their production is costly and laborious. Moreover, there are also other challenges related to immunogenicity, carcinogenicity, low target cell specificity and inability to transfer large size genes. On the other hand, non-viral vectors can be relatively easy and cost-effectively produced. They are safe, can transfer large size genes and are less toxic. Their main disadvantages are low transfection efficiency and poor transgene expression [48]. Having considered the above, in this group, only the Sleeping Beauty (SB) transposon/transposase system with clustered regularly interspaced short palindromic repeats (CRISP/Cas9) has great potential [49]. Table 1 below lists the characteristics of different engineering methods of CARs.

A CAR intervention example of a mechanism in patients is shown in Figure 2.

The SB transposon system requires only two components: transposon DNA and transposase enzyme [57]. The most efficient way to deliver selected components into the target cell is the classical two-plasmid configuration: one for SB transposase and other for an artificial transposon flanked via terminal inverted repeats (TIR) on both sides of a vector [58]. This system also takes into account the origin of replication component and antibiotic resistance gene of choice. To setup transposon into the target cell, transfection or electroporation can be used [57]. To eliminate toxicity effect and decrease the immunogenic reaction of DNA transfection, it is best to use the current state-of-the-art delivery methods, messengers mRNA or minicircle DNA (MC) [59]. The SB’s production disadvantages are poor protein stability, low solubility and aggregation properties; however, incorporation of two mutations I212S and C176S into SB100X transposase improves these features [60]. A recent study indicates that using a catalytically inactive Cas9 (dCas9) with single-guide RNA approach may facilitate genetic material insertion into a genome [61]. Cas9 is a dual RNA-guided DNA endonuclease enzyme that uses base pairing to recognise and cleave target DNA with complementarity to the guide RNA, such as invading bacteriophage DNA or plasmid DNA [62]. Tethering the transposase toward a target that is overexpressed in the human genome dramatically increases the number of possible attach points and thus induces chances of targeted transposition with a flexible and easy-to-use RNA-guided system [63]. Pilot studies have indicated that SB is a safe and effective tool to manufacture therapeutic CAR-T cells in cancers [64,65,66,67].

## 4. In Vitro and In Vivo Studies

In recent years, CARs proved to be particularly effective in patients with haematological cancers [68,69]. Solid tumours, however, remain challenging because of their histopathological structure, aberrant vasculature and extensive vascular leakage [70]. In the OC (solid malignancies) therapy with the CAR-T cells, the key issues are lack of target antigen specificity, intrinsic target antigen heterogeneity, an immunosuppressive TME, expression of immune checkpoint molecules, ineffective intracellular trafficking/infiltration and low persistence.

The critical issue related to CAR-T-cell therapy in solid tumours is the identification of corresponding tumour target antigens absent or expressed at remarkably low levels in healthy tissue, most notably in vital organs. This problem is further amplified because each particular CAR-T cell only needs to recognise a few receptors on the target cell for full activation to occur. Selecting an ideal target antigen (i.e., overexpressed on tumour cells and with minimal or no expression on healthy tissues) will eliminate off-target effects and associated toxicity [71,72,73]. Yet another issue relates to the immunosuppressive TME. TME contains various interacting components, including tumour cells, immune cells, stromal cells, chemokines, cytokines and extracellular matrix. In solid tumours, TME exhibits strong immunosuppressive effects due to the recruitment of tumour-associated macrophages (TAMs), cancer-associated fibroblasts (CAFs), myeloid-derived suppressor cells (MDSCs) and regulatory T cells (Treg), and the production of immunosuppressive cytokines and soluble factors (e.g., IL-10, VEGF, TGFβ, indoleamine 2,3-dioxygenase and adenosine) [74].

A hypoxic, low pH intrinsic microenvironment and the activated inhibitory pathways appear to be problematic when it comes to T-cell trafficking and T-cell infiltration into tumour sites [75]. These adaptive survival-oriented cancer changes contribute to the induction of selectively enhanced permeability and retention of lipid particles and macromolecular substances. Therefore, in a search for a successful solid tumour therapy using CARs the specific cell surface antigens identification is crucial. To reduce ‘on-target, off-tumour toxicity’, an idea of introducing a regulated suicide gene into CARs such as the HSV-TK (herpes simplex virus I–derived thymidine kinase) or iCasp9 (caspase 9) has been proposed. Both, the T cells with HSV-TK and iCasp9 genes prevent alloreactivity, exhibit low potential immunogenicity and no acute toxicity without compromising their functional and phenotypic characteristics [76].

The cell surface antigens targeted by CARs include proteins, carbohydrates and glycolipids. In in vivo and in vitro studies most common antigens targeted by CARs in OC cells include MUC16, folate receptor-α (FRα), mesothelin and HER2 (Table 2).

Mesothelin (MSLN), a cell surface glycoprotein, is generally expressed in mesothelial cells lining the pleura, peritoneum (minimally on the epithelial cells of the ovaries and fallopian tubes) and pericardium, however highly expressed in many tumour cells, including OC; its soluble form can also be found in the bloodstream of OC patients [90]. Studies have identified MSLN as a promising tumour antigen in OC as it is overexpressed in over 75% of HGSOC tumours [91]. A number of agents including CAR T-cells targeting MSLN have been developed, and are currently being investigated. There were also other preclinical studies conducted focusing on the use of mesothelin-based CAR-T cells in subcutaneous or in situ mouse models of mesothelioma, ovarian cancer and lung cancer transplantation [77,78,79].

Recent studies reveal that annexin 2 (ANXA2) has been detected in OC. Overexpression of ANXA2 mediates extracellular matrix degradation and neovascularisation by the production of plasmin and correlates with invasion and metastasis [80]. Lately, it has been suggested that natural killer (NK) cells may be better chimeric antigen receptor drivers than T cells because of their favourable innate features, such as direct recognition and elimination of tumour cells [79]. To overcome ‘on-target off-tumour’ cytotoxicity, the dual-target CARs may be a better choice [92]. It has been shown that dual CARs are related to the longer survival time of mice up to two times when compared to single CAR groups and control group (80 vs. 40 days) [83].

The alpha isoform, folate receptor α (FRα), also known as gene *FOLR1* or folate binding protein (FBP), is a glycosylphosphatidylinositol (GPI)-anchored membrane protein that binds folic acid with high affinity and transports folate (vitamin B9) by receptor-mediated endocytosis. FRα has been reported to be overexpressed in solid tumours such as OC, precisely 90%, but has restricted expression in normal cells. From the perspective of OC, where increasing levels of tissue FR are associated with tumour progression, it is an attractive therapeutic target [93]. Moreover, FRα expression is not affected by any earlier treatment attempts using chemotherapy. Having considered that, folate receptor α is ideal for a tumour antigen in targeted treatments of OC [81]. The first team that constructed CAR-T cells targeting FRα and used the CAR-T cells to treat OC was Kershaw et al.—the murine MOv18 scFv was used and a signalling domain of the Fc receptor γ chain [40]. It was later demonstrated that CAR-T cells targeting FRα is safe to administer, despite not showing the desired therapeutic effects. Ao at al. further verified that the anti-FRα CARs redirect NK-92 cells with specific anti-tumour activity, and the third-generation anti-FRα CAR-engineered NK-92 cells display more potent cytotoxicity against FRα-positive OC [81]. Song et al. investigated the coupling of the FRα-specific site scFv (MOv19) with the T-cell receptor CD3ζ chain signalling module alone (MOv19-ζ) or in combination with the CD137 (4-1BB) costimulatory motif in tandem (MOv19-BBζ). In the co-culture process of FRα(+) OC cells, MOv19-ζ and MOv19-BBζ may result in increased secretion of various inflammatory factors, such as IFN-γ, IL-2, TNF-α and IL-4. Moreover, in intraperitoneal, subcutaneous and lung metastases in FRα(+) animal models, the use of MOv19-BBζ CAR-T cells have shown positive therapeutic effects [86].

CXCR1 (interleukin-8 receptor alpha [IL-8RA]) is the G protein-coupled receptor that binds IL-8 with high affinity. The proinflammatory cytokine IL-8 expression produced by tumour tissues to recruit leukocytes is substantially higher in a wide range of tumour types, as well as in OC [94,95]. Whilding et al. showed that IL-8 is actually produced by many αvβ6-positive cancer cell lines, among them SKOV3, and is present in the circulation of mice engrafted with various tumour xenografts expressing this integrin [96]. It has been reported that circulating IL-8 levels correlate with disease severity and prognosis in a number of solid tumours, where it is involved in a wide range of pathological functions, including angiogenesis, support of tumour stem cells survival and immunosuppressive myeloid cells recruitment [97]. CXCR1- and CXCR2-containing CAR-T cells showed increased migration towards IL-8 and conditioned media containing this chemokine. Furthermore, T cells that co-expressed CAR A20-28z and CXCR2 increased tumour control in vivo compared to CAR T cells deprived of this chemokine receptor, without accompanying toxicity [96]. Ng et al. demonstrated that expression of the IL-8 receptor CXCR1 to match CAR-NK cells to a chemokine secreted by the tumour facilitated increased migration and infiltration into the tumour and improved the anti-tumour responses of the immune effector cells in vivo [82]. Having considered various studies results, it seems that PD-1 is another ideal target for CAR T therapy. Programmed cell death-1 (PD-1), also called programmed cell death-ligand 1 inhibitor, is an immune checkpoint immunomodulator highly expressed on antigen-presenting cells, hepatocytes and tumours. Interaction with programmed cell death-1 results in inhibition of antigen-specific responses on T cells, B cells and macrophages. PD-1 belongs to the CD28/cytotoxic T lymphocyte-associated antigen-4(CTLA-4) family [98]. Antibodies that block PD-1/PD-L1 interaction reduce signalling between co-inhibitory molecules. It is also known that T cells are able to secrete cytokines, such as IL-10 and IFN-γ, to induce the generation of a CTLA ligand on OC cells, e.g., PD-1. At the same time, PD-1 induces expression and binds to inhibitory receptors on the surface of T cells. This reduces the anti-activity of effector T cells and directs T cells’ movement to sites of inflammation. Sometimes, it results in T cells being unable to avoid the immune response [99]. Yet another experiment (on mice with melanoma) revealed that the T cells escape from immune surveillance is suppressed after upregulation of the PDL-1 expression in the tumour microenvironment (TME). The T-cell infiltration could, however, be made considerably greater. In order to achieve that, intraperitoneal injection of a PD-1 antibody is required as this procedure aims to block the PD-1 pathway [100]. Another study where patients with low PDL-1 expression were compared to patients with high PDL-1 expression revealed that the five-year survival rate is considerably greater in the former group [101].

From a clinical perspective, the most crucial peripheral checkpoint inhibitor pathway exploited by tumour cells within the TME identified to date is the interaction between the PD-1 receptor on T cells with its programmed death-ligand 1 (PD-L1) and programmed death-ligand 2 (PD-L2) on tumour cells. Increased expression of PD-L1 on T4 CAR T cells occurred when these cells were in culture with OC cells. By contrast, EOC cell lines exhibited increased PD-L1 expression after chemotherapy treatment [102].

A particular class of targets that has had limited exploration in CARs against solid tumours are glycoepitopes. External glycosylation in cancer can be initiated via dysregulation of glycosyltransferases, altering both the function and molecular profile of tumour cells. It is generally agreed that abnormal glycosylation of tumour cells leads to creation of new connections with immune cells that actively suppress anti-tumour immunity. Therefore, as tumour-specific glycosylation patterns determine the immune suppressive nature of tumours, their interactions with endogenous carbohydrate-binding proteins (lectins) could be considered as new immune checkpoints to be targeted by immunotherapy [103].

Mucin 16 (MUC16, cancer antigen 125, CA125) is mainly overexpressed in ovarian cancer (above 80%) with the shedding of antigens in a soluble form or membrane-bound form that can suppress humoral immunity, especially antibody-dependent cytotoxicity (ADCC). CA125, encoded by MUC16, is a well-known circulating marker of early stage disease that is monitored in the clinical course of OC patients. MUC16 is a macromolecule transmembrane mucin consisting of a single membrane-spanning domain, a cytoplasmic tail, an extensive N-terminal domain and a tandem repeat sequence, with CA125 antigen in the MUC16 tandem repeat. The interaction between MUC16 and MSLN contributes to peritumoral adhesion and spheroid formation, thus providing a targeting strategy that is being developed to reduce peritumoral metastasis and facilitate other therapies [104]. MUC16-CAR-T cells injected intravenously or intraperitoneally are able to delay OC’s progression or altogether remove tumours in mouse tumour-bearing models. Therefore, again it seems that MUC16 is an ideal antigenic target for CAR molecules [83].

L1 cell adhesion molecule (L1-CAM) is a 200–220 kDa transmembrane glycoprotein of the immunoglobulin (Ig) superfamily. It plays a vital role in neuronal cell adhesion and migration, such as neurite outgrowth guidance, axon binding, myelination, synaptogenesis and long-term potentiation. The abnormal expression of L1-CAM protein is strongly correlated with the aggressive behaviour of many human malignancies. Mechanistic studies showed that forcibly altered L1-CAM expression significantly alters cell properties, including invasion, migration, proliferation and chemoresistance [105]. Hong et al. have shown that the L1-CAM is highly over-expressed in ovarian cancer, while absent in normal ovaries [84], and that its expression on tumours is also associated with poor clinical outcome [106]. The same team demonstrated that L1-CAM-specific CAR T cells allow considerable control of solid tumour growth in an in vivo ovarian cancer xenograft model that exhibited clinically significant manifestations of widespread tumour metastasis in the peritoneal cavity and massive ascites.

Human epidermal growth factor receptor 2 (HER2; also called Her-2/neu or ErbB2) is a member of the transmembrane epidermal growth factor receptor family and is one of the most studied TAAs for cancer immunotherapy. HER2 is a proto-oncogene and plays a vital role in the pathogenesis and clinical process of various tumours. In vitro and animal experiments have clearly shown that gene amplification and protein overexpression of HER2/neu play a key role in tumorigenic transformation and development of tumours [107]. Subsequent studies have shown that HER2/neu gene amplification and overexpression are associated with OC while protein expression in normal tissues is negative or very low. Overexpressed HER2/neu proteins make tumours more aggressive and are independent risk factors for poor prognosis in these cancer patients [108]. Sun et al. constructed and evaluated a novel anti-HER2 chA21 scFv-based CAR. The results of this study show that novel chA21 scFv-based, HER2-specific CAR T cells not only recognised and killed HER2+ breast and ovarian cancer cells ex vivo but also induced regression of experimental breast cancer in vivo. The data support further exploration of the HER2 CAR T-cell therapy for HER2-expressing cancers [85]. At present, HER2-specific CAR-T-cell therapy has shown good therapeutic potential in the preclinical stage. However, HER2-CAR-T-cell treatment in OC is still in the clinical experimental stage.

The follicle-stimulating hormone receptor (FSHR) is thought to be selectively expressed in women in ovarian granulosa cells and at low levels in the ovarian endothelium. This surface antigen is expressed in 50–70% of serous OCs, although its expression in other histological types of OC remains unknown. Perales-Puchalt et al. revealed that in immunocompetent mice growing syngeneic, orthotopic and aggressive ovarian tumours, fully murine FSHR-targeted T cells increased the survival without any measurable toxicity. In that study, chimeric receptors enhanced endogenous tumour-reactive T cells’ ability to abrogate malignant progression upon adoptive transfer into naive recipients subsequently challenged with the same tumour [87].

There is a promising solution to prevent systemic toxicity—it requires to combine tumour-specific protein, e.g., NKG2D linked to IL-2. Interleukin-2 is a cytokine from the cytokine-receptor γ-chain family with many potentially useful functions including stimulation of T cells, NK cells and immunoglobulins [109]. NKG2D is a transmembrane protein belonging to the NKG2 family of C-type lectin-like receptors. The NKG2D proteins are stress-induced self-proteins entirely absent or present only at low levels on normal cells’ surface. Still, they become overexpressed by infected, transformed, senescent and stressed cells [110]. Kang et al. showed that TC-1 tumour-bearing mice treated with a therapeutic HPV type 16E7 DNA vaccine and then given the DNA construct encoding the chimeric NKG2D-Fc-IL2 protein demonstrated reduced tumour mass growth and prolonged survival. Specific delivery of IL-2 with the NKG2D-Fc system led to the expansion of tumour antigen-specific CD8+ T cells at the tumour loci and an improved therapeutic anti-tumour effect generated by the therapeutic DNA HPV vaccine [111]. Other molecules that combine with the NKG2D-Fc system could be IL-12, IL-15 or GM-CSF [112].

Wang et al. designed a novel anti-uPAR CAR consisting of antigen recognition domain using a natural amino-terminal fragment, a part of the A chain of uPA instead of scFv to construct the third-generation CAR (ATF-CAR) T cells against OC cells in vitro [113]. uPAR (urokinase plasminogen activator receptor) is a receptor for uPA involved in the conversion of plasminogen to plasmin, which degrades the extracellular matrix (ECM) during tumour migration and metastasis. uPAR also affects other signals which induce tumorigenesis, tumour proliferation and adhesion and tumour dormancy and reactivation in OC [114]. It is worth mentioning that uPAR expression in healthy cells is relatively rare and focuses upon healing and tissue remodelling process and inflammatory response in some macrophages, endothelial cells and respiratory cells, which make this receptor an excellence choice for CAR development [115]. At a ratio of 10:1 ATF-CAR T cells exhibited significant lysis cytotoxicity against uPAR-positive cells SKOV3, HO8910, C13K and ES-2. Moreover, they were shown to produce higher levels of Th1 cytokines [113].

The 5T4 oncofoetal antigen was first identified during a search for surface molecules shared between human trophoblasts and cancer cells with the rationale that they may function to allow survival of the foetus as a semiallograft in the mother or a tumour in its host. The 5T4 is a 72-kDa transmembrane protein expressed on the placenta and a wide range of human carcinomas [116]. The 5T4 is known to be highly expressed in OC, and its expression correlates with more advanced stages of disease (FIGO stages III and IV) and with poorly differentiated tumours. Patients whose tumours express 5T4 seem to have a worse progression-free and overall survival [117]. Owens et al. has shown that polyclonal lymphocytes isolated from the peripheral blood of patients with OC, can be redirected to target tumour cells expressing 5T4 effectively. Co-culture of CAR T cells with matched autologous tumour disaggregates resulted in antigen-specific secretion of IFN-γ. Assessment of anti-5T4 CAR T cells’ efficacy in a mouse model allowed to discover a therapeutic benefit against the established ovarian tumours [88].

Several researchers attempted to combine CAR intervention with other therapies. Wahba et al. showed that in vivo paclitaxel synergises with ErbB-targeted CAR T cells (T4) [102]. The ErbB family of proteins contains four receptor tyrosine kinases, structurally related to the epidermal growth factor receptor (EGFR) and via PI3-K/AKT pathway leads to increased cell proliferation and inhibition of apoptosis. Paclitaxel binds to the h-tubulin subunit and stabilises the microtubules, resulting in disruption of normal microtubule dynamics during cell division. Failure of microtubule separation during the G2/M phase blocks cell mitosis and results in apoptosis. DNA damage caused by chemotherapy leads to cleavage and activation of intracellular caspases, initiating a proteolytic cascade and eventually cell death. Reversal of apoptosis can be achieved using the pan-caspase inhibitor carbobenzoxy-valyl-alanyl-aspartyl-[O-methyl]-fluoromethylketone (Z-VAD), which binds to caspase proteases irreversibly, preventing the initiation of the proteolytic cascade. In their study treating ovarian tumour cells with chemotherapy and Z-VAD resulted in a reversal of the anti-tumour activity observed following chemotherapy treatment. When Z-VAD was used with chemotherapy and T4 cells, there was a partial, yet significant reversal in the reduction seen in tumour cell viability. The reversal was not complete, suggesting that caspase induction, or indeed apoptosis, was not the sole mechanism but was definitely contributing to the combination therapy’s synergistic effect. Mannose-6-phosphate receptor-mediated autophagy and the arrest of the cell cycle in G2/M have also shown to be induced by chemotherapy and significantly contributing to the synergy [102].

## 5. Clinical Trials

Using CARs has resulted in successful outcomes in hematopoietic malignancies and inspired introduction of similar strategies to treat solid tumours [118,119,120]. Despite encouraging results of in vitro and in vivo studies, the solid cancer methods of treatment are not developed enough to achieve the desired results. There are a few studies that describe the potential application of CARs in the treatment strategy of patients with OC. In early clinical trials of first-generation CAR T cells for OC, safety and therapeutic efficacy were difficult to be determined because of the aforementioned poor in vivo expansion and persistence of the transferred lymphocytes. For example, Wright et al. investigated whether mucin 1 variable number tandem repeat (VNTR)-stimulated mononuclear cells (M1SMC) can be given safely intraperitoneally to subjects with recurrent OC after resection and chemotherapy [121]. In the study, 7 participants underwent up to 4 cycles of treatment. Each time patients were subjected to leukapheresis (separation of white blood cells from a blood sample) before intraperitoneal infusion of tumour-specific cytotoxic T-lymphocytes. There was no other intervention performed on the subjects. The therapy was well tolerated; the only clinical side effect was abdominal pain in one patient. Median survival was 11.5 months; one subject was free of disease at the end of the study. After the first month of immunotherapy, the tumour marker CA-125 was not significantly reduced from the statistical point of view. Nevertheless, after that time, its significance increased. The killer cells, cytokine production and memory T-lymphocytes increased after the first cycle of stimulation but plateaued or decreased after that. The percent of NK cells inversely correlated with other immune parameters [121]. Unlike other tumours which do not typically possess physical barriers that would prevent their interactions with CAR T cells, many OC tumours have formidable barriers that render these masses inaccessible to invasion by immune cells.

The next clinical trial involved 15 patients, and the treatment was based on lentiviral-transduced chimeric antigen receptor (CAR)-modified autologous T cells redirected against mesothelin (CART-meso) cells (single infusion 1–3 × 10^7^–10^8^ cells/m^2^ i.v.). The most common adverse events were low-grade fatigue and nausea observed in 47% and 40% of the group. Lymphodepletion improved the initial expansion of CART-meso cells but did not impact CART-meso cell persistence. However, researchers detected CAR DNA in tumour biopsies and ascites from several patients, suggesting CART-meso have infiltrating abilities. A single infusion of CART-meso cells was safe in this human study but produced minimal anti-tumour activity. The best overall response was stable disease (11/15 patients) [122]. Studies evaluating a fully human anti-mesothelin and other CARs in OC treatment are ongoing (Table 3).

Challenges such as the immunosuppressive character of the TME, CAR-T cell persistence and trafficking to the tumour seem to limit CAR-T-cell efficacy in solid cancers [123]. Over the past decade, significant efforts have been made to develop CARs targeting OC. Comprehensive descriptions of promising CAR candidates have recently been published [33,79,80,81,97,124]. Because of the immunosuppressive cells in the TME, including tumour-associated macrophages (TAMs), myeloid-derived suppressor cells (MDSCs) and regulatory T cells (Tregs), the anti-tumour immune function of OC patients is significantly attenuated. Thus, patients have the poorest outcomes after receiving immunotherapy. Numerous CAR strategies to affect the TME have been proposed, these were either directly aiming at cell surface components supporting the tumour, or combining the tumour-targeted CAR with anti-immuno-inhibitory drugs such as checkpoint inhibitors [124].

Cancer stem cells as a new target for CARsAs results of various studies suggest a notable phenomenon in the typical clinical course of OC is stem cell-driven repopulation. From the perspective of our research and the role of cancer stem cells (CSCs) in developing and advancing solid tumour malignancies, we can suggest that this disease is particularly well-suited for this purpose. Three theories describe the origin of CSCs: a normal stem cell, transit-amplifying cell, or a normal progenitor cell [125]. CSCs are precursors and protoplasts of heterogeneous mature cancer cells with a huge capacity for self-renewal and are believed to be a key factor for tumour development and recurrence. It is well-known that CSC cells are resistant to chemotherapy and radiotherapy. In view of this fact, it seems worthwhile to focus on immunotherapy. According to many confirmed clinical data, it has been advocated as a promising strategy to control problematic CSC cells [30,126].

CSC cells, similarly to tissue stem cells (TSC), have an enhanced ability to resist harmful internal and external factors. Due to an excellent ability to repair DNA damage, resistance to low proliferation rates, upregulation of detoxification enzymes and efflux pumps, CSC cells are much less sensitive to the use of radiotherapy, cytotoxic drugs or targeted therapies. Because CSCs seek to mimic the function of their healthy counterparts, in preliminary studies researchers used the same techniques, such as ALDH1s’s detoxifying enzymatic activity, that were used to define stem cells in putative tissues of origin, particularly the fallopian tube and ovaries. In contrast, other studies have focused on surface proteins. These proteins’ expression has previously been demonstrated on cells with stem cell properties in other types of cancer, such as CD24, CD44 and CD133, to identify CSCs in OC [127].

Ovarian cancer stem cells (OCSC)s were identified only a few years ago. Parte et al. have investigated 34 samples of OC human tissue and revealed the presence and different distribution of various CSC surface markers (CD133, CD24 and CD44), functional CSC markers (ALDH1/2) and cell proliferation (KI67) specific markers [127]. Klapdor et al. developed a novel anti-CD24 CAR targeting OC. This new 3rd generation anti-CD24 CAR has been shown to be highly specific and exhibits powerful cytotoxic activity against CD24-positive OC cell lines and primary cells [78]. Also, specific elimination of ovarian CSCs by anti-CD133-CAR expressing NK92 cells offers quite an optimistic strategy that, if confirmed in vivo, should form the foundation of future clinical research aimed at preventing relapses [128].

Collected data suggest that many surface markers present on CSCs surface (CD133, CD44, CD47) could be used as a target for chimeric antigen receptors. Note, that markers must be selected with exceptional due care and be tumour-specific, not expressed via healthy cells, as it could result in severe side effects. Embellishing or arming CARs to be switchable (toxicity controlled by antigen dosage) could be a method of choice. Irrespective of this, OCSCs exhibit significant phenotypic and functional heterogeneity, which is vital in designing and developing targeted therapies. For this reason, it is necessary to conduct research studies on a regular basis to explain all the heterogeneity features of CSCs in OC. The same applies to the challenge of determining their association with histopathological subtypes, clinical parameters and molecular aberrations. Given the recent advances in the analysis of single cells at the genetic level, transcriptome and proteome profiling, it seems we have finally amassed enough tools and knowledge to address this issue.

## Figures and Tables

**Figure 1 ijms-22-03495-f001:**
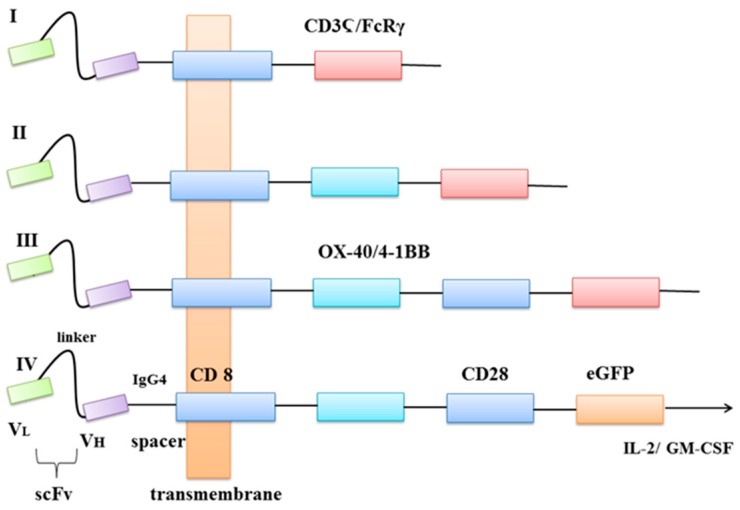
Four generations of CARs. V_L_—light chain variable domain, V_H_—heavy chain variable domain, scFv—a single-chain variable fragment, spacer—protein fragments fused together, CD8—transmembrane protein, OX-40—also known as CD134 glycoprotein receptor, tumour necrosis factor receptor superfamily, 4-1BB—glycoprotein receptor tumour necrosis factor receptor superfamily, CD3Ϛ—protein complex and T-cell co-receptor that is involved in activating both the cytotoxic T cell and T helper cells, FcRγ—receptor for inducing phagocytosis, CD28—a protein that provides costimulatory signals, eGFP—enhanced green fluorescent protein, Il-2—interleukin 2 (cytokine), GM-CSF—granulocyte-macrophage colony-stimulating factor.

**Figure 2 ijms-22-03495-f002:**
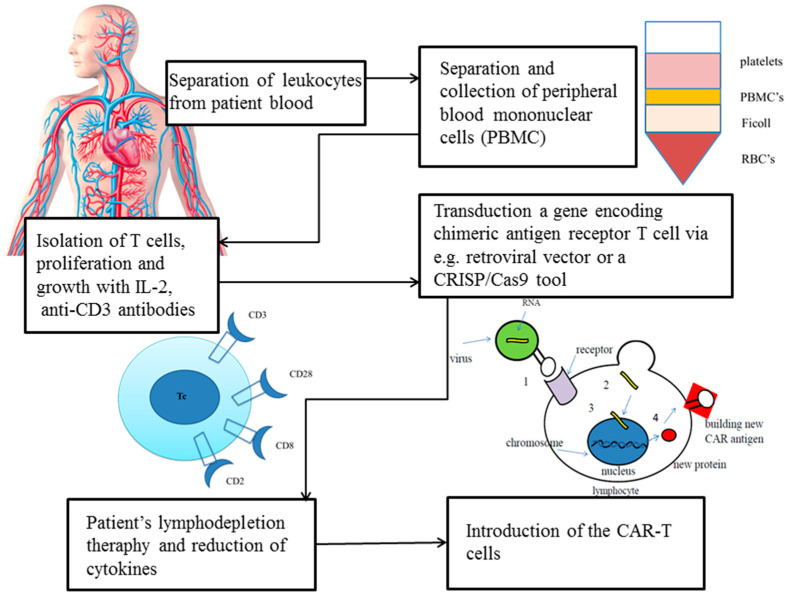
Transduction of a gene encoding a CAR in T cell via a retroviral vector or a CRISP/Cas9 tool [56].

**Table 1 ijms-22-03495-t001:** The characteristic of different engineering methods of CARs.

Genetic Approach	Methods	Structure	Study	Target	Advantages	Disadvantages	Source
**Viral**	Retroviral vectors	ssRNA	in vivo	only mitosis	substitutability↑	insertional mutagenesis↑ titre vector production ↑	[50]
	Lentiviral vectors	ssRNA	in vivo	entire cycle	integration ↑ risk of insertional mutagenesis ↓	possible insertional mutagenesis↑ presence of regulatory proteins in the packaging construct transient expression of the transgene with integration-defective vector↑	[51]
	Adenoviruses	dsDNA	in vivo	entire cycle	toxicity↓	integrity↓	[52]
	Adeno-associated viral vectors	ssDNA	in vivo	entire cycle	infection efficiency ↑ gene expression ↑	internalisation ↓ endosomal trafficking↓ nuclear import↓	[53]
**Nonviral**	Liposome-mediated gene transfer	lipid n-layer	in vitro	entire cycle	condensation of DNA ↑ infection efficiency ↑	transfection efficiency↓	[54]
	Messenger RNA-mediated gene transduction	ssRNA	in vitro	entire cycle	insertional mutations ↓ potential malignant transformation/ genotoxicity ↓ off-tumour, on-target side effects ↓	instable, non-biocompatible↓ low biodegradability, low efficacy↓ toxicity at high dose, difficult preparation, low transformation efficiency↓	[46,55]
	Sleeping Beauty transposon/transposase system	plasmid-plasmid	in vivo	entire cycle	integration ↑	insertional mutagenesis ↓	[50]

**Table 2 ijms-22-03495-t002:** Antigenic targets being exploited for CAR-T-cell therapy in OC treatment.

Target Antigen	Cells	Gene Transfer	Intervention/ Monitoring	Study	Studied Material	Outcomes	Source
MSLN	T/NK	messenger RNA-mediated gene transduction	i.p. 1 × 10^8^ cells/ up to 6 weeks	in vivo	Defb29 Vegf-luc/Hmeso Platinum resistant OC	↓tumour burdens ↑mice survival	[77]
CD24 MSLN	NK-29	lentiviral transduction	5 × 10^4^ cells/ 24 h	in vitro	A2780, OVCAR3, SKOV3, Primary OC	↑cytotoxicity	[78]
MSLN	NK	transposon vector	i.p. 1.5 × 10^7^ cells/ up to 7 weeks	in vivo	A1847, MA 148	↑inhibition of tumour growth ↑survival	[79]
ANXA2	T	lentiviral vector	i.p. 5 × 10^6^ cells/ up to 5 days	in vivo	IGROV-1, SKOV3	↑cytokine release ↑cytotoxicity ↑survival ↓tumour burdens	[80]
FRα	NK	lentiviral vector	i.p. 1 × 10^6^ cells/ up to 10 days	in vivo	SKOV3, A2780, HTC116, A431	↑elimination of cancer cells ↑survival	[81]
CXCR1	NK	mRNA transfection	i.v. 5 × 10^6^ cells/twice a week/2 weeks	in vivo	SKOV3, CaOV3, SW626	ascites generation↓ ↓tumour cells in ascites samples complete metabolic response ↑survival	[82]
PDL1 MUC16	T	lentiviral infection	i.p. 1 × 10^6^ cells/ up to 4 weeks	in vivo	SKOV3	↑IL-2, IFN-γ, TNF-α ↑regression of ovarian cells ↑survival	[83]
L1-CAM	Tcm	lentiviral vector	i.p. 5 × 10^6^/up to 17 weeks	in vivo	CAOV-3, OVCAR-3, SKOV-3, MADH2780, A2780	regression of tumours in the peritoneal cavity and massive ascites	[84]
HER2/ neu	T	lentiviral vector	1 × 10^5^ CAR/1 × 10^5^ tumour cells	in vitro	SKOV3, OVCAR3, A2780, A1847	↑expression of CARs ↑cytotoxicity ↓tumour cells	[85]
FRα	T	retroviral vector	i.v. up to 5 × 10^5^ cells/48 h	in vivo	14 patients with recurrent, resected recurrent, or residual epithelial FR+ ovarian cancer	The treatment was well tolerated, but no antitumour effect was observed.	[40]
FRα	T	lentiviral vector	i.v. up to 1 × 10^6^ cells/4 weeks	in vivo	SKOV-3, OVCAR3, A1847, C30, PEO-1	tumour regression T-cell persistence↓	[86]
FSHR	T	retroviral vector	i.p. 2 injections up to 1.5 × 10^6^ cells/up to 50 days	in vivo	mouse xenografts OVCAR-3, CaOV3, RNG1, OVTOKO and TOV-21G	increased survival no toxicity	[87]
5T4	T	lentiviral vector	i.p. up to 6 × 10^4^ cells/100 days	in vivo	SKOV-3	5T4-specific CAR can recognise and respond physiologically to autologous tumour cells	[88]
TAG72		lentiviral vector	i.p.,i.v. 5 × 10^6^ cells/up to 8 weeks	in vivo	mouse xenografts SKOV-3, OVCAR-3, OVCAR-3	↓tumour growth ↑survival	[89]

MSLN-mesothelin, CD24—signal transducer sialoglycoprotein, ANXA2—annexin 2, FRα—folate receptor α, CXCR1—chemokine receptor 1, PDL1—programmed death-ligand 1, MUC16—mucin 16, L1-CAM—L1 transmembrane protein family, HER2/neu—receptor tyrosine-protein kinase erbB-2, FSHR—follicle-stimulating hormone receptor, 5T4—trophoblast glycoprotein (TPBG), TAG72—tumour-associated glycoprotein 72.

**Table 3 ijms-22-03495-t003:** Actually running studies including the application of CAR-T chimeric antigen receptors in solid cancer treatment, according to ClinicalTrials.gov.

Study Title	Summary	Intervention	Phase	Locations
The Fourth Generation CART-cell Therapy for Refractory-Relapsed OC	The goal of this clinical trial is to study the safety and feasibility of anti-Mesothelin Chimeric Antigen Receptor T-Cell (MESO CAR-T cells) therapy for Refractory-Relapsed OC	Autologous genetically modified anti-MESO CAR transduced T cells	Early 1	Shanghai 6th People’s Hospital Shanghai, China
Safety and Effectiveness of MESO-CAR T-Cells Therapy for Relapsed and Refractory Epithelial Ovarian Cancer	The goal of this clinical trial is to study the feasibility and efficacy of anti-MESO antigen receptors (CARs) T-cell therapy for relapsed and refractory epithelial ovarian cancer	Retroviral vector-transduced autologous T cells to express anti-MESO CARs Fludarabine 30 mg/m^2^/d Cyclophosphamide 300 mg/m^2^/d	1 and 2	The Second Affiliated hospital of Zhejiang University School of Medicine Hangzhou, China
A Clinical Trial of MESO-CAR T-Cells Therapy for Relapsed and Refractory Ovarian Cancer MESO-CAR T Cells	The goal of this clinical trial is to study the feasibility and efficacy of anti-MESO antigen receptors (CARs) T-cell therapy for relapsed and refractory ovarian cancer	Retroviral vector-transduced autologous T cells to express anti-MESO CARs Fludarabine 30 mg/m^2^/d; Cyclophosphamide 300 mg/m^2^/d	Early 1	The Second Affiliated hospital of Zhejiang University School of Medicine Hangzhou, China
A Single-Center, Phase I Clinical Study to Evaluate the Safety, Tolerability and Efficacy of LCAR-M23, a CAR-T-Cell Therapy Targeting MSLN in Patients With Relapsed and Refractory Epithelial Ovarian Cancer	This study is a prospective, single-arm, single-centre, open-label, single-dose dose finding and extension study to evaluate the safety, tolerability, pharmacokinetics and anti-tumour efficacy profiles of the LCAR-M23 CAR-T-cell therapy in subjects with relapsed and refractory epithelial ovarian cancer after prior adequate standard of care	LCAR-M23 cells Prior to infusion of LCAR-M23, subjects will receive a premedication regimen (IV of cyclophosphamide 300 mg/m^2^ and fludarabine 30 mg/m^2^ once daily for 3 days)	1	Shanghai East Hospital Shanghai, China
A Single-Arm, Single-Center, Open-Label Pilot Study of Anti-ALPP CART-cells in Patient With Alkaline Phosphatase, Placental (ALPP)-Positive Metastatic Ovarian and Endometrial Cancer	The goal of this clinical trial is to evaluate the safety and efficacy of anti-ALPP chimeric antigen receptor (CAR)-modified T (CAR-T) cells in treating patients with ALPP-positive metastatic ovarian and endometrial cancer.	CART treatment Retroviral vector-transduced autologous T cells to express anti-ALPP CARs Cyclophosphamide will be administered at dose of 20 mg/kg for 1 day and then fludarabine will be given for the next 3 days with 35 mg/m^2^ and then the CAR-T cells will be administered	1 and 2	Xinqiao Hospital of Chongqing’ China
An Exploratory Study of αPD1-MSLN-CAR T Cells Secreting PD-1 Nanobodies for the Treatment of MSLN-positive Advanced Solid Tumours	This is a single arm, open-label, dose escalation clinical study to evaluate the safety and tolerability of autologous mesothelin (MSLN)-targeted chimeric antigen receptor (MSLN-CAR) T cells secreting PD-1 nanobodies (αPD1-MSLN-CAR T cells) in patients with solid tumours	αPD1-MSLN-CAR T cells Subjects will undergo leukapheresis to isolate peripheral blood mononuclear cells (PBMCs) for the production of αPD1-MSLN-CAR T cells. The initial dose of 1 × 10^5^ CAR+ T cells/kg will be infused on day 0.	Early 1	Shanghai Tenth people’s Hospital Shanghai, China
Phase I Study Evaluating Benefit of PRGN-3005 UltraCAR-T™ (Autologous CAR T Cells) in Advanced Stage Platinum Resistant Ovarian Cancer Patients	This is a study to identify the best dose and side effects of modified immune cells PRGN-3005 (autologous chimeric antigen receptor (CAR) T cells developed by Precigen, Inc) in treating patients with ovarian, fallopian tube, or primary peritoneal cancer that has spread to other places in the body, that has come back and is resistant to platinum chemotherapy.	PRGN-3005 UltraCAR-T cells given IP or IV	1	Fred Hutch/University of Washington Cancer Consortium Seattle, United States
A Phase 1 Study of Autologous Activated T-cells Targeting the B7-H3 Antigen in Subjects With Recurrent Epithelial Ovarian Cancer	This is single centre, open-label phase 1 dose escalation trial that uses modified 3+3 design to identify a recommended phase 2 dose (RP2D) of CAR.B7-H3 T cell. An expansion cohort will enrol additional subjects at the RP2D for a total enrolment of up to 21 subjects on the protocol.	CAR.B7-H3 Two dose levels will be evaluated: Dose Level 1 (7.5 × 10^7^ cells/infusion), dose Level 2 (2 × 10^8^ cells/infusion).	1	Lineberger Comprehensive Cancer Center Chapel Hill, United States
Phase I Clinical Trial of Adoptive Transfer of Autologous Folate Receptor-Alpha Redirected T Cells for Recurrent High Grade Serous Ovarian, Fallopian Tube, or Primary Peritoneal Cancer	Phase I study to establish safety and feasibility of IP(L) administered lentiviral transduced MOv19-BBz CAR T cells with or without cyclophosphamide + fludarabine as lymphodepleting chemotherapy.	MOv19-BBz CAR T cells IP administered lentiviral transduced MOv19-BBz CAR T cells with or without cyclophosphamide + fludarabine as lymphodepleting chemotherapy	1	University of Pennsylvania Health System Philadelphia, United States
Innovative Treatment of Ovarian Cancer Based on Immunogene-modified T Cells (IgT)	The primary objectives are to evaluate the safety and efficacy of infusion of autologous OC immunogene-modified T cells (OC-IgT cells)	OC-IgT cells. Autologous human OC-IgT cells	1 and 2	Shenzhen Geno-immune Medical Institute Shenzhen, China
A Phase 1 Open-Label, Multi-Center First in Human Study of TnMUC1-Targeted Genetically-Modified Chimeric Antigen Receptor T Cells in Patients With Advanced TnMUC1-Positive Solid Tumours and Multiple Myeloma	Phase 1 study of the safety, tolerability, feasibility and preliminary efficacy of the administration of genetically modified autologous T cells (CART-TnMUC1 cells) engineered to express a chimeric antigen receptor (CAR) capable of recognizing the tumour antigen, TnMUC1 and activating the T cell (CART- TnMUC1 cells)	CART-TnMUC1 Single IV administration of genetically modified autologous T cells engineered to express a TnMUC1-Targeted Genetically-Modified Chimeric Antigen (CAR) Drug: Cyclophosphamide	1	The Angeles Clinic and Research Institute Los Angeles and 7 others, United States
Autologous Immunotherapy With Multi-target Gene-modified CAR-T/TCR-T Cell for Malignancies	This is a single arm, open-label, uni-center, phase I-II study to evaluate the safety and effectiveness of CAR-T/TCR-T-cell immunotherapy in treating with different malignancies patients (OC and 13 more)	CAR-T-cell immunotherapy According to tumour burden and other conditions, patients will be treated with cyclophosphamide or fludarabine, then, CAR-T. cells will be infused 48-72 h later	1 and 2	The First Affiliated Hospital of Zhengzhou University Zhengzhou, China
Phase I Study of Human Chimeric Antigen Receptor Modified T Cells in Patients With Mesothelin Expressing Cancers	Phase I study to establish safety and feasibility of IV or IP(L) administered lentiviral transduced huCART-meso cells with or without lymphodepletion by way of administering cyclophosphamide	huCART-meso cells IV or IP(L) lentiviral transduced huCART-meso cells in 6 cohorts with and without cyclophosphamide in a 3+3 dose escalation design.	1	University of Pennsylvania Philadelphia, United States
A Phase I Trial to Assess Safety, Tolerability and Anti-tumour Activity of Autologous T Cell Modified Chimeric Antigen Receptor (CAR) (CCT303-406) in Patients With Relapsed or Refractory HER2 Positive Solid Tumours	This clinical study is to investigate the safety and tolerability of CCT303-406 CAR modified autologous T cells (CCT303-406) in subjects with relapsed or refractory stage IV metastatic HER2-positive solid tumours	CCT303-406 Blood will be collected from subjects to isolate peripheral blood mononuclear cells for the production of CCT303-406. cyclophosphamide and fludarabine for lymphodepletion followed by a single dose of CCT303-406 via IV.	1	Zhongshan Hospital Affiliated to Fudan University Shanghai, China

BW-body weight, IP—intraperitoneal, IP(L)—intrapleural, IV—intravenous, TC—tumour cells.

## Data Availability

Not applicable.

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
