# Peer review of "Chimeric Antigen Receptor Design and Efficacy in Ovarian Cancer Treatment"

_ijms, 2021, doi:10.3390/ijms22073495_

Round 1
Reviewer 1 Report
The authors of the article have carried out a review of the role of Chimeric Antigen Receptor in ovarian cancer treatment. There are numerous reviews about this emerging immunotherapy technique, focused especially in hematological malignancies. the article is not very original since there are several recent review articles about Chimeric Antigen Receptor in ovarian cancer. Some of these are not included
Yan W, Hu H, Tang B. Advances Of Chimeric Antigen Receptor T Cell Therapy In Ovarian Cancer. Onco Targets Ther. 2019 Sep 30;12:8015-8022. doi: 10.2147/OTT.S203550.
Jindal V, Arora E, Gupta S, Lal A, Masab M, Potdar R. Prospects of chimeric antigen receptor T cell therapy in ovarian cancer. Med Oncol. 2018 Apr 12;35(5):70. doi: 10.1007/s12032-018-1131-6.
This year an article very similar to this has been published:
Marofi F, Motavalli R, Safonov VA, Thangavelu L, Yumashev AV, Alexander M, Shomali N, Chartrand MS, Pathak Y, Jarahian M, Izadi S, Hassanzadeh A, Shirafkan N, Tahmasebi S, Khiavi FM. CAR T cells in solid tumors: challenges and opportunities. Stem Cell Res Ther. 2021 Jan 25;12(1):81.
Although it is not focused on ovarian cancer, it has a specific section for this type of cancer.
In table 1 in the Messenger RNA-mediated gene transduction section, the article of reference 46, must be included: Rajan, T.S.; Gugliandolo, A.; Bramanti, P.; Mazzon, E. In Vitro-Transcribed mRNA Chimeric Antigen Receptor T Cell (IVT mRNA CAR T) Therapy in Hematologic and Solid Tumor Management: A Preclinical Update. Int. J. Mol. Sci. 2020, 21, 6514.This article is more recent and more directly related to mRNA-mediated gene traducci
Table 3 includes the application of chimeric antigen receptors in cancer treatment, according to ClinicalTrials.gov. but on this website there are more interesting studies not shown in this table for example, CAR T Cells in Mesothelin Expressing Cancers and CAR-macrophages for the Treatment of HER2 Overexpressing Solid Tumors. Both studies include ovarian cancer.
The section 6, Cancer stem cells as a new target for CARs it is not well developed, the information is minimal and confusing. There are better reviews on this topic, for example Wang, L.; Xu, T.; Cui, M. Are ovarian cancer stem cells the target for innovative immunotherapy? Onco Targets Ther. 2018, 11, 2615-2626. Most of the information presented in this section belongs to this article, the rest of the articles used are not specific to stem cells.
Some paragraphs are too similar to other articles. thus lines 294-309 of this article are very similar to a paragraph of the introduction of Li, T.; Wang, J. Therapeutic effect of dual CAR-T targeting PDL1 and MUC16 antigens on ovarian cancer cells in mice. BMC Cancer 2020, 20, 678. I include both paragraphs so that they can be compared.
Katarzyna Terlikowska, Bożena Dobrzycka and Sławomir Terlikowski. Chimeric Antigen Receptor Design and Efficacy in Ovarian Cancer Treatment
Programmed cell death-1 (PD-1) is an immune checkpoint receptor highly expressed by activated T cells, B cells, antigen-presenting cells and macrophages. PD-1 belongs to the CD28/cytotoxic T lymphocyte-associated antigen-4(CTLA-4) family [95]. This immunosuppressive molecule and its ligand PDL1 form the PD1/PDL1 signalling pathway, playing an inhibitory role in T cell immunity. The most recent study suggests that T cells are able to secrete cytokines, such as IL-10 and IFN-γ, to induce the generation of a CTLA ligand on OC cells, e.g., PD-1. Simultaneously, PD-1 induces expression and binds to inhibitory receptors on T cells’ surface, subsequently reducing the anti-activity of effector T cells and directing T cell repositioning or causing T cells to fail in escaping immune response [96]. An experiment involving mice with melanoma revealed that up-regulation of the PDL-1 expression in the tumour microenvironment leads to suppressing anti-tumour immune escape by T cells, however, the T cell infiltration can be significantly increased upon intraperitoneal injection of a PD-1 antibody in order to block the PD-1 pathway [97]. Another study showed that the five-year survival rate of patients with low PDL- 1 expression is considerably higher than patients with high PDL-1 expression [98]. Having considered results of the above studies, it seems that PD-1 is yet another ideal target for CAR T therapy.
Li, T.; Wang, J. Therapeutic effect of dual CAR-T targeting PDL1 and MUC16 antigens on ovarian cancer cells in mice. BMCCancer 2020, 20, 678. doi: 10.1186/s12885-020-07180-x.
Programmed cell death-1(PD1) is an immunosuppressive molecule widely expressed on the surface of activated T cells, B cells, antigen-presenting cells, and macrophages. It belongs to the CD28/cytotoxic T lymphocyte-associated antigen-4(CTLA-4) family [15]. PD-1 and its ligand PDL1 constitute the PD1/PDL1 signaling pathway, which plays an inhibitory role in T cell immunity. Current research suggests that T cells can secrete cytokines such as IL-10 and IFN-γ to induce generation of CTLA ligand, such as PD1 expressing on OC cells. At the same time, PD1 induces expression and combines with inhibitory receptors on the surface of T cells, therefore reducing the anti-activity of effector T cells and guiding T cell reposition or causing T cell failure to achieve immune escape [16–20]. In the experiment of melanoma-bearing mice, it was found that the up-regulated expression of PDL1 in the tumor microenvironment led to the suppression of anti-tumor immune escape on T cells. After intraperitoneal injection of the PD1 antibody to block the PD1 pathway, T cell significantly increased infiltration [21–23]. There is also a study that shows the five-year survival rate of patients with low expression of PDL1 is significantly higher than that of patients with high expression of PDL1 [22, 24, 25]. Based on the above research of PD1, we surmise that PDL1 would be another ideal target.
Editing errors are present in the submitted manuscript
Table 1 source
Table 2 study, source
Table 3 Recruiting
Lines 137-144 text format
Lines 475-784 text format
Lines 522-529 text format
Abreviations ANNXN2, MUC16, PDL1/L2… and more
Author Response
Comments and Suggestions for Authors
The authors of the article have carried out a review of the role of Chimeric Antigen Receptor in ovarian cancer treatment. There are numerous reviews about this emerging immunotherapy technique, focused especially in hematological malignancies. the article is not very original since there are several recent review articles about Chimeric Antigen Receptor in ovarian cancer. Some of these are not included
Yan W, Hu H, Tang B. Advances Of Chimeric Antigen Receptor T Cell Therapy In Ovarian Cancer. Onco Targets Ther. 2019 Sep 30;12:8015-8022. doi: 10.2147/OTT.S203550.
Jindal V, Arora E, Gupta S, Lal A, Masab M, Potdar R. Prospects of chimeric antigen receptor T cell therapy in ovarian cancer. Med Oncol. 2018 Apr 12;35(5):70. doi: 10.1007/s12032-018-1131-6.
This year an article very similar to this has been published:
Marofi F, Motavalli R, Safonov VA, Thangavelu L, Yumashev AV, Alexander M, Shomali N, Chartrand MS, Pathak Y, Jarahian M, Izadi S, Hassanzadeh A, Shirafkan N, Tahmasebi S, Khiavi FM. CAR T cells in solid tumors: challenges and opportunities. Stem Cell Res Ther. 2021 Jan 25;12(1):81.
Although it is not focused on ovarian cancer, it has a specific section for this type of cancer.
Re: The above articles have been added: references 118, 119, 120 verse 450.
In table 1 in the Messenger RNA-mediated gene transduction section, the article of reference 46, must be included: Rajan, T.S.; Gugliandolo, A.; Bramanti, P.; Mazzon, E. In Vitro-Transcribed mRNA Chimeric Antigen Receptor T Cell (IVT mRNA CAR T) Therapy in Hematologic and Solid Tumor Management: A Preclinical Update. Int. J. Mol. Sci. 2020, 21, 6514.This article is more recent and more directly related to mRNA-mediated gene traducci
Re: Article of reference 46 was added in Table 1.
Table 3 includes the application of chimeric antigen receptors in cancer treatment, according to ClinicalTrials.gov. but on this website there are more interesting studies not shown in this table for example, CAR T Cells in Mesothelin Expressing Cancers and CAR-macrophages for the Treatment of HER2 Overexpressing Solid Tumors. Both studies include ovarian cancer.
Re: Table 3 has been completed.
The section 6, Cancer stem cells as a new target for CARs it is not well developed, the information is minimal and confusing. There are better reviews on this topic, for example Wang, L.; Xu, T.; Cui, M. Are ovarian cancer stem cells the target for innovative immunotherapy? Onco Targets Ther. 2018, 11, 2615-2626. Most of the information presented in this section belongs to this article, the rest of the articles used are not specific to stem cells.
Some paragraphs are too similar to other articles. thus lines 294-309 of this article are very similar to a paragraph of the introduction of Li, T.; Wang, J. Therapeutic effect of dual CAR-T targeting PDL1 and MUC16 antigens on ovarian cancer cells in mice. BMC Cancer 2020, 20, 678. I include both paragraphs so that they can be compared.
Re: Sections 6 have been redrafted: verse 502-545.
Katarzyna Terlikowska, Bożena Dobrzycka and Sławomir Terlikowski. Chimeric Antigen Receptor Design and Efficacy in Ovarian Cancer Treatment
Programmed cell death-1 (PD-1) is an immune checkpoint receptor highly expressed by activated T cells, B cells, antigen-presenting cells and macrophages. PD-1 belongs to the CD28/cytotoxic T lymphocyte-associated antigen-4(CTLA-4) family [95]. This immunosuppressive molecule and its ligand PDL1 form the PD1/PDL1 signalling pathway, playing an inhibitory role in T cell immunity. The most recent study suggests that T cells are able to secrete cytokines, such as IL-10 and IFN-γ, to induce the generation of a CTLA ligand on OC cells, e.g., PD-1. Simultaneously, PD-1 induces expression and binds to inhibitory receptors on T cells’ surface, subsequently reducing the anti-activity of effector T cells and directing T cell repositioning or causing T cells to fail in escaping immune response [96]. An experiment involving mice with melanoma revealed that up-regulation of the PDL-1 expression in the tumour microenvironment leads to suppressing anti-tumour immune escape by T cells, however, the T cell infiltration can be significantly increased upon intraperitoneal injection of a PD-1 antibody in order to block the PD-1 pathway [97]. Another study showed that the five-year survival rate of patients with low PDL- 1 expression is considerably higher than patients with high PDL-1 expression [98]. Having considered results of the above studies, it seems that PD-1 is yet another ideal target for CAR T therapy.
Li, T.; Wang, J. Therapeutic effect of dual CAR-T targeting PDL1 and MUC16 antigens on ovarian cancer cells in mice. BMCCancer 2020, 20, 678. doi: 10.1186/s12885-020-07180-x.
Programmed cell death-1(PD1) is an immunosuppressive molecule widely expressed on the surface of activated T cells, B cells, antigen-presenting cells, and macrophages. It belongs to the CD28/cytotoxic T lymphocyte-associated antigen-4(CTLA-4) family [15]. PD-1 and its ligand PDL1 constitute the PD1/PDL1 signaling pathway, which plays an inhibitory role in T cell immunity. Current research suggests that T cells can secrete cytokines such as IL-10 and IFN-γ to induce generation of CTLA ligand, such as PD1 expressing on OC cells. At the same time, PD1 induces expression and combines with inhibitory receptors on the surface of T cells, therefore reducing the anti-activity of effector T cells and guiding T cell reposition or causing T cell failure to achieve immune escape [16–20]. In the experiment of melanoma-bearing mice, it was found that the up-regulated expression of PDL1 in the tumor microenvironment led to the suppression of anti-tumor immune escape on T cells. After intraperitoneal injection of the PD1 antibody to block the PD1 pathway, T cell significantly increased infiltration [21–23]. There is also a study that shows the five-year survival rate of patients with low expression of PDL1 is significantly higher than that of patients with high expression of PDL1 [22, 24, 25]. Based on the above research of PD1, we surmise that PDL1 would be another ideal target.
Re: The above paragraph has been redrafted: verse 301-320.
Editing errors are present in the submitted manuscript
Table 1 source
Table 2 study, source
Table 3 Recruiting
Lines 137-144 text format
Lines 475-784 text format
Lines 522-529 text format
Abreviations ANNXN2, MUC16, PDL1/L2… and more
Re: Editing errors were caused by editorial formatting and have been corrected
Reviewer 2 Report
Important review article highlighting the great potential of using CART cells to treat ovarian cancer
Minor corrections
Correction in abstract abbreviation for annexin A2 should be ANXA2
Add ref 116 to table 2
Add ref 106 to table 2
Add following ref to table 2 ‘Murad JP, Kozlowska AK, Lee HJ, Ramamurthy M, Chang WC, Yazaki P, Colcher D, Shively J, Cristea M, Forman SJ, Priceman SJ. Effective Targeting of TAG72(+) Peritoneal 20 Ovarian Tumors via Regional Delivery of CAR-Engineered T Cells. Front Immunol 2018;9: 2268. PMID: 30510550’
Add ref 90 to table 2
Add ref 40 to table
Check if other refs mentioned in the text are not included in Table 2
Author Response
Comments and Suggestions for Authors
Important review article highlighting the great potential of using CART cells to treat ovarian cancer
Minor corrections
Correction in abstract abbreviation for annexin A2 should be ANXA2
Re: The abbreviation was corrected-verse 22
Add ref 116 to table 2
Add ref 106 to table 2
Add following ref to table 2 ‘Murad JP, Kozlowska AK, Lee HJ, Ramamurthy M, Chang WC, Yazaki P, Colcher D, Shively J, Cristea M, Forman SJ, Priceman SJ. Effective Targeting of TAG72(+) Peritoneal 20 Ovarian Tumors via Regional Delivery of CAR-Engineered T Cells. Front Immunol 2018;9: 2268. PMID: 30510550’
Add ref 90 to table 2
Add ref 40 to table
Check if other refs mentioned in the text are not included in Table 2
Re: The indicated references have been added to Table 2.
Comments and Suggestions for Authors
Important review article highlighting the great potential of using CART cells to treat ovarian cancer
Minor corrections
Correction in abstract abbreviation for annexin A2 should be ANXA2
Re: The abbreviation was corrected-verse 22
Add ref 116 to table 2
Add ref 106 to table 2
Add following ref to table 2 ‘Murad JP, Kozlowska AK, Lee HJ, Ramamurthy M, Chang WC, Yazaki P, Colcher D, Shively J, Cristea M, Forman SJ, Priceman SJ. Effective Targeting of TAG72(+) Peritoneal 20 Ovarian Tumors via Regional Delivery of CAR-Engineered T Cells. Front Immunol 2018;9: 2268. PMID: 30510550’
Add ref 90 to table 2
Add ref 40 to table
Check if other refs mentioned in the text are not included in Table 2
Re: The indicated references have been added to Table 2.
Comments and Suggestions for Authors
Important review article highlighting the great potential of using CART cells to treat ovarian cancer
Minor corrections
Correction in abstract abbreviation for annexin A2 should be ANXA2
Re: The abbreviation was corrected-verse 22
Add ref 116 to table 2
Add ref 106 to table 2
Add following ref to table 2 ‘Murad JP, Kozlowska AK, Lee HJ, Ramamurthy M, Chang WC, Yazaki P, Colcher D, Shively J, Cristea M, Forman SJ, Priceman SJ. Effective Targeting of TAG72(+) Peritoneal 20 Ovarian Tumors via Regional Delivery of CAR-Engineered T Cells. Front Immunol 2018;9: 2268. PMID: 30510550’
Add ref 90 to table 2
Add ref 40 to table
Check if other refs mentioned in the text are not included in Table 2
Re: The indicated references have been added to Table 2.
Comments and Suggestions for Authors
Important review article highlighting the great potential of using CART cells to treat ovarian cancer
Minor corrections
Correction in abstract abbreviation for annexin A2 should be ANXA2
Re: The abbreviation was corrected-verse 22
Add ref 116 to table 2
Add ref 106 to table 2
Add following ref to table 2 ‘Murad JP, Kozlowska AK, Lee HJ, Ramamurthy M, Chang WC, Yazaki P, Colcher D, Shively J, Cristea M, Forman SJ, Priceman SJ. Effective Targeting of TAG72(+) Peritoneal 20 Ovarian Tumors via Regional Delivery of CAR-Engineered T Cells. Front Immunol 2018;9: 2268. PMID: 30510550’
Add ref 90 to table 2
Add ref 40 to table
Check if other refs mentioned in the text are not included in Table 2
Re: The indicated references have been added to Table 2.
Comments and Suggestions for Authors
Important review article highlighting the great potential of using CART cells to treat ovarian cancer
Minor corrections
Correction in abstract abbreviation for annexin A2 should be ANXA2
Re: The abbreviation was corrected-verse 22
Add ref 116 to table 2
Add ref 106 to table 2
Add following ref to table 2 ‘Murad JP, Kozlowska AK, Lee HJ, Ramamurthy M, Chang WC, Yazaki P, Colcher D, Shively J, Cristea M, Forman SJ, Priceman SJ. Effective Targeting of TAG72(+) Peritoneal 20 Ovarian Tumors via Regional Delivery of CAR-Engineered T Cells. Front Immunol 2018;9: 2268. PMID: 30510550’
Add ref 90 to table 2
Add ref 40 to table
Check if other refs mentioned in the text are not included in Table 2
Re: The indicated references have been added to Table 2.
Reviewer 3 Report
The authors present an interesting review article on CARs for ovarian cancer. This field has already been covered by other reviews. however, this review includes important information which merits publication.
I have a few comments that should be considered:
Introduction:
The introduction is too long. The authors should keep the focus on the CAR therapy rather than discussing too much therapeutic strategies for ovarian cancer. The first two paragraphs should be united in one. The paragraph on genetic mutations could be deleted. It does not give any important information for the topic of the review.
Figure 1: Why include eGFP in vector IV? The varying style regarding the descriptions is really confusing. Would prefer a more standardized presentation.
Table 1: It would be interesting to add, if vectors have been used in vivo (add references). Additional to structure would be good to add size of DNA. Please add disadvantages for lentiviral and mRNA. There is no arrow after nuclear import
Figure 2: This figure is way too schematic. Would be great to add at least some schematic illustrations that help to understand the strategy.
Table 2: The title is misleading. This table does not show characteristics of different targets. it shows selected characteristics of published studies. But it is not complete at all. So, either add more studies and targets or report why you selected this group of studies.
Table 3: Please add targets, engineering method, early results. Do you only want to include actually running studies or also already conducted ones? Then they should be included.
Author Response
The authors present an interesting review article on CARs for ovarian cancer. This field has already been covered by other reviews. however, this review includes important information which merits publication.
I have a few comments that should be considered:
Introduction:
The introduction is too long. The authors should keep the focus on the CAR therapy rather than discussing too much therapeutic strategies for ovarian cancer. The first two paragraphs should be united in one. The paragraph on genetic mutations could be deleted. It does not give any important information for the topic of the review.
Re: The first two paragraphs have been merged. Genetic mutation verse deleted.
Figure 1: Why include eGFP in vector IV? The varying style regarding the descriptions is really confusing. Would prefer a more standardized presentation.
Re: The green fluorescent protein (GFP) is a protein that exhibits bright green fluorescence when exposed to light in the blue to ultraviolet range. It can be added to the every generation of CAR in term to estimate its specificity to bind target antigen via fluorescence microscope. – verse 31-34
Table 1: It would be interesting to add, if vectors have been used in vivo (add references). Additional to structure would be good to add size of DNA. Please add disadvantages for lentiviral and mRNA. There is no arrow after nuclear import
Re: In vitro or in vivo study and disadvantages for lentiviral and mRNA was added.
Figure 2: This figure is way too schematic. Would be great to add at least some schematic illustrations that help to understand the strategy.
Re: Figure 2 has been corrected.
Table 2: The title is misleading. This table does not show characteristics of different targets. it shows selected characteristics of published studies. But it is not complete at all. So, either add more studies and targets or report why you selected this group of studies.
Re: The title of Table 2 has been changed: Antigenic targets being exploited for CAR-T cell therapy in OC treatment. Four studies have been added.
Table 3: Please add targets, engineering method, early results. Do you only want to include actually running studies or also already conducted ones? Then they should be included.
Re: The title of Table 3 has been changed. Targets and engineering methods have been added.
The authors present an interesting review article on CARs for ovarian cancer. This field has already been covered by other reviews. however, this review includes important information which merits publication.
I have a few comments that should be considered:
Introduction:
The introduction is too long. The authors should keep the focus on the CAR therapy rather than discussing too much therapeutic strategies for ovarian cancer. The first two paragraphs should be united in one. The paragraph on genetic mutations could be deleted. It does not give any important information for the topic of the review.
Re: The first two paragraphs have been merged. Genetic mutation verse deleted.
Figure 1: Why include eGFP in vector IV? The varying style regarding the descriptions is really confusing. Would prefer a more standardized presentation.
Re: The green fluorescent protein (GFP) is a protein that exhibits bright green fluorescence when exposed to light in the blue to ultraviolet range. It can be added to the every generation of CAR in term to estimate its specificity to bind target antigen via fluorescence microscope. – verse 31-34
Table 1: It would be interesting to add, if vectors have been used in vivo (add references). Additional to structure would be good to add size of DNA. Please add disadvantages for lentiviral and mRNA. There is no arrow after nuclear import
Re: In vitro or in vivo study and disadvantages for lentiviral and mRNA was added.
Figure 2: This figure is way too schematic. Would be great to add at least some schematic illustrations that help to understand the strategy.
Re: Figure 2 has been corrected.
Table 2: The title is misleading. This table does not show characteristics of different targets. it shows selected characteristics of published studies. But it is not complete at all. So, either add more studies and targets or report why you selected this group of studies.
Re: The title of Table 2 has been changed: Antigenic targets being exploited for CAR-T cell therapy in OC treatment. Four studies have been added.
Table 3: Please add targets, engineering method, early results. Do you only want to include actually running studies or also already conducted ones? Then they should be included.
Re: The title of Table 3 has been changed. Targets and engineering methods have been added.
The authors present an interesting review article on CARs for ovarian cancer. This field has already been covered by other reviews. however, this review includes important information which merits publication.
I have a few comments that should be considered:
Introduction:
The introduction is too long. The authors should keep the focus on the CAR therapy rather than discussing too much therapeutic strategies for ovarian cancer. The first two paragraphs should be united in one. The paragraph on genetic mutations could be deleted. It does not give any important information for the topic of the review.
Re: The first two paragraphs have been merged. Genetic mutation verse deleted.
Figure 1: Why include eGFP in vector IV? The varying style regarding the descriptions is really confusing. Would prefer a more standardized presentation.
Re: The green fluorescent protein (GFP) is a protein that exhibits bright green fluorescence when exposed to light in the blue to ultraviolet range. It can be added to the every generation of CAR in term to estimate its specificity to bind target antigen via fluorescence microscope. – verse 31-34
Table 1: It would be interesting to add, if vectors have been used in vivo (add references). Additional to structure would be good to add size of DNA. Please add disadvantages for lentiviral and mRNA. There is no arrow after nuclear import
Re: In vitro or in vivo study and disadvantages for lentiviral and mRNA was added.
Figure 2: This figure is way too schematic. Would be great to add at least some schematic illustrations that help to understand the strategy.
Re: Figure 2 has been corrected.
Table 2: The title is misleading. This table does not show characteristics of different targets. it shows selected characteristics of published studies. But it is not complete at all. So, either add more studies and targets or report why you selected this group of studies.
Re: The title of Table 2 has been changed: Antigenic targets being exploited for CAR-T cell therapy in OC treatment. Four studies have been added.
Table 3: Please add targets, engineering method, early results. Do you only want to include actually running studies or also already conducted ones? Then they should be included.
Re: The title of Table 3 has been changed. Targets and engineering methods have been added.
The authors present an interesting review article on CARs for ovarian cancer. This field has already been covered by other reviews. however, this review includes important information which merits publication.
I have a few comments that should be considered:
Introduction:
The introduction is too long. The authors should keep the focus on the CAR therapy rather than discussing too much therapeutic strategies for ovarian cancer. The first two paragraphs should be united in one. The paragraph on genetic mutations could be deleted. It does not give any important information for the topic of the review.
Re: The first two paragraphs have been merged. Genetic mutation verse deleted.
Figure 1: Why include eGFP in vector IV? The varying style regarding the descriptions is really confusing. Would prefer a more standardized presentation.
Re: The green fluorescent protein (GFP) is a protein that exhibits bright green fluorescence when exposed to light in the blue to ultraviolet range. It can be added to the every generation of CAR in term to estimate its specificity to bind target antigen via fluorescence microscope. – verse 31-34
Table 1: It would be interesting to add, if vectors have been used in vivo (add references). Additional to structure would be good to add size of DNA. Please add disadvantages for lentiviral and mRNA. There is no arrow after nuclear import
Re: In vitro or in vivo study and disadvantages for lentiviral and mRNA was added.
Figure 2: This figure is way too schematic. Would be great to add at least some schematic illustrations that help to understand the strategy.
Re: Figure 2 has been corrected.
Table 2: The title is misleading. This table does not show characteristics of different targets. it shows selected characteristics of published studies. But it is not complete at all. So, either add more studies and targets or report why you selected this group of studies.
Re: The title of Table 2 has been changed: Antigenic targets being exploited for CAR-T cell therapy in OC treatment. Four studies have been added.
Table 3: Please add targets, engineering method, early results. Do you only want to include actually running studies or also already conducted ones? Then they should be included.
Re: The title of Table 3 has been changed. Targets and engineering methods have been added.
The authors present an interesting review article on CARs for ovarian cancer. This field has already been covered by other reviews. however, this review includes important information which merits publication.
I have a few comments that should be considered:
Introduction:
The introduction is too long. The authors should keep the focus on the CAR therapy rather than discussing too much therapeutic strategies for ovarian cancer. The first two paragraphs should be united in one. The paragraph on genetic mutations could be deleted. It does not give any important information for the topic of the review.
Re: The first two paragraphs have been merged. Genetic mutation verse deleted.
Figure 1: Why include eGFP in vector IV? The varying style regarding the descriptions is really confusing. Would prefer a more standardized presentation.
Re: The green fluorescent protein (GFP) is a protein that exhibits bright green fluorescence when exposed to light in the blue to ultraviolet range. It can be added to the every generation of CAR in term to estimate its specificity to bind target antigen via fluorescence microscope. – verse 31-34
Table 1: It would be interesting to add, if vectors have been used in vivo (add references). Additional to structure would be good to add size of DNA. Please add disadvantages for lentiviral and mRNA. There is no arrow after nuclear import
Re: In vitro or in vivo study and disadvantages for lentiviral and mRNA was added.
Figure 2: This figure is way too schematic. Would be great to add at least some schematic illustrations that help to understand the strategy.
Re: Figure 2 has been corrected.
Table 2: The title is misleading. This table does not show characteristics of different targets. it shows selected characteristics of published studies. But it is not complete at all. So, either add more studies and targets or report why you selected this group of studies.
Re: The title of Table 2 has been changed: Antigenic targets being exploited for CAR-T cell therapy in OC treatment. Four studies have been added.
Table 3: Please add targets, engineering method, early results. Do you only want to include actually running studies or also already conducted ones? Then they should be included.
Re: The title of Table 3 has been changed. Targets and engineering methods have been added.